# Hypergamy reconsidered: Marriage in England, 1837–2021

Gregory Clark[1,2,3,4], Neil Cummins [3,4]*

1 Department of Economics, University of Southern Denmark, Odense, Syddanmark, Denmark, 2 Danish Institute for Advanced Study (DIAS), Odense, Syddanmark, Denmark, 3 Department of Economic History, London School of Economics and Political Science, London, United Kingdom, 4 Center for Economic Policy Research (CEPR), Paris, France

* n.j.cummins@lse.ac.uk

## Abstract

It is widely believed that women value social status in marital partners more than men, leading to female marital hypergamy ("marrying up"), and more female intergenerational social mobility. Using evidence from more than 33 million marriages and 67 million births in England 1837–2021 we show that within this era there was never significant hypergamous marriage by women. The average status of women's fathers equaled that of their husbands' fathers. For marriages 1912–2007 the average social status of female surnames equaled that of male. This was true also for parent surnames of children. Consistent with this, there was no differential tendency in England of men and women to marry by family status. There is also ancillary evidence that physical attraction cannot have been the significant determinant of matching in marriages in any period 1837–2021, based on the very strong correlation observed in underlying social status for marital partners throughout these years.

## Introduction

Studies of partner preferences in marriage find differences in desired mate characteristics between men and women. Women prize social status in the mate more highly, and men prize physical attributes, including youth [1–9]. It is believed that these differences in mate preferences are longstanding, and led to systematic female marital hypergamy, even in recent years [10, 11]. Women, it is claimed, typically married men who were older and of higher social status than their own families. Correspondingly men married women who were younger and of lower social status. On average women "married up", and men "married down."

In this paper we examine the extent of hypergamy for marriages in England 1837–2021 using two new sources. The first is a database of 1.7 million church marriages in England 1837–2021, which records the occupations of the fathers of both bride and groom. Here we can measure with great accuracy the extent of hypergamy by period 1837–2021. The second source is population register data on marriages and births in England by surname, 1912–2007. Since for rarer surnames we can assign an average status by surname, based on house value by surname in 1999, we can test whether average surname status of grooms with rare surnames exceeded that of brides with rare surnames. Further, given the rising importance of non-

**Data Availability Statement:** The data may be accessed at the following URL: https://www.openicpsr.org/openicpsr/project/213682/version/V1/view.

**Funding:** The author(s) received no specific funding for this work.

**Competing interests:** The authors have declared that no competing interests exist.

marital unions, we can also use the birth register to test whether children on average had fathers of higher surname status than for their mothers.

With these databases we show, for England, the following:

1. There is no significant hypergamy by women in English marriage throughout 1837–2021, as measured by bride and groom fathers' average occupational status, or by bride and groom average surname house value.

2. A pattern which will produce hypergamy is where high status men are more likely to marry than low status men, while high status women are less likely to marry than low status women. Across the family status distribution male and female relative marriage rates were always the same. There is no differential tendency to marry across family status for women compared to men.

3. Women show no more social mobility in their marital pairings than do men. Across the parent status distribution, women generally match to men in just the same way as men match to women.

4. There is ancillary evidence that in England 1837–2021 the physical appearance of women was a modest determinant of matching in marriage. The underlying correlation in social abilities was high and constant at 0.8 1837–2021. Such a high correlation would not be possible if men valued physical appearance in women strongly in marriage, and physical appearance was largely uncorrelated with social abilities.

## Methods

In this paper we measure hypergamy by the relative status of the bride and groom birth families. Most of the literature on this topic has focused on hypergamy measured as the relative educational or income status of bride and groom themselves [12–23]. Since the relative educational levels of women have increased greatly in the last 200 years, this will imply strong female hypergamy in earlier years, and a recent move to women having equal or higher status than their spouses. Such a measure of hypergamy, however, will not accurately reflect the relative family social class of marital partners in earlier years. Such a measure will also not reveal whether women exhibit more social mobility than men through marriage. This is why we measure hypergamy here using family background.

The literature on hypergamy in Western societies, measured using the family background of the partners, is much sparser. There is widespread assumption in the anthropology, psychology, and sociology literatures that hypergamy of this type was and is common social practice [1, 3–6, 10, 24–29]. But there is no systematic demonstration of its existence other than a recent paper on partnering in Norway [11]. For England there seem to be no studies showing the prevalence of female hypergamy in this sense, either historically or now. In the US there are such studies, but they find no evidence of hypergamy. Rubin (1968), for example, finds no evidence for hypergamy, measured by father occupational status, for a sample of 26,000 women born 1900–1940 [30]. Similarly Charles et al. (2012) find no difference for husbands and wives in the US surveyed in 1988, in the parental wealth of husbands versus wives [31].

In order for the marital preference differences by gender discussed above to produce on average hypergamous marriages for women, there have to be ancillary conditions other than a straight pairing of all men with all women in marriage. A one-to-one pairing of all men and women would entail that the family status of men and women on average would be equal in marriage, no matter gender difference in partner preferences.

For the average female marriage to be hypergamous we need conditions such as low status men being less likely to marry, and/or high status women being less likely to marry. In some earlier societies this outcome was created by a surplus of males in the population and also polygyny, where all women married but lower status men were excluded. Thus pre-industrial Chinese demographic and marriage patterns have been argued to produce female hypergamy. While all women married, there was a surplus of males from female infanticide, so that low status men were not able to find brides. Exacerbating this shortage of brides for poorer men was the practice of richer men of taking multiple wives, or concubines. Hypergamy was the norm [32].

In England, the equal numbers of men and women, and also monogamy, restrict the possibilities for hypergamy. But we could still get hypergamy if the unmarried are drawn from the top of the female family status distribution, and from the bottom of the male. This is the outcome Almås et al. (2023), detect for modern Norway where they find that female partnering frequencies relative to male are higher for lower family-status women. Almås et al. count people as partnered both if they are formally married, but also if they are registered as joint parents of a child [11]. We can check using the English marriage and birth registers whether in England either of these groups, the married and parents, display different male and female marriage rates relative to social status.

To measure the average difference in family social status between grooms and brides in England 1837–2021, we employ the following sources.

## Marriage database, 1837–2021

From 1837–2024 a marriage certificate in England and Wales, includes: (1) marriage date and place (2) names of the bride and the groom, their ages, their marital condition (single/divorced/widowed), and their "rank or profession" (3) names and "rank or profession" of their fathers (4) signatures or marks of the bride and groom.

The UK government now has such records of around 106 million marriages 1837–2024 from England and Wales. However, it costs £11 to obtain a copy of any marriage certificate from the government. Since copies of the marriage certificates were kept in church registers, and many of these registers have since been deposited in local record offices, these church marriages provide a convenient alternative source for marriage records.

The marriage certificates available in local record offices exclude Civil Marriages in registry offices. But Civil marriages remained a minority of all weddings before 1914. In 1841 they were 1.7% of all marriages, and in 1914, still only 24%. Thereafter there were increasing numbers of civil weddings, as church attendance declined, but also as divorce rates increased. Until recently divorcees were rarely granted permission to be remarried in the Church of England. 31% of weddings were civil by 1952, and 68% by 2012 [33].

Thus Church marriage records will give an good picture of the hypergamous nature of marriage overall 1837–1914. But there will be potential bias from the increasing omission of civil marriages as we go from 1914 to 2021. As described in the S1 File we obtained from amateur genealogists nearly 1.7 million transcripts of such church marriage records, mainly, however, from the years before 1940. For the years 1940–2021 we supplemented these records with a set of 27,887 marriage records from Essex, where Essex conveniently includes both parts of what is now London, as well as rural areas.

For marriages 1940–2021 we assigned occupational scores to fathers using the CAMSIS 1990 scores of social status [34]. These scores are derived as maximizing the similarity of father-son occupational scores [35]. For marriages 1837–1939 we constructed our own occupational status association index constructed from the father-son and father-in-law-son

occupational pairings to also maximize intergenerational occupational status correlations. This is a similar method to that applied in the well-known HISCAM index, but with our much greater set of data we derived a more accurate status index than HISCAM [36]. Clark, Cummins and Curtis (2024) details the construction of this new index, and why it performs better than the HISCAM index for Britain 1800–1938 [37].

We employ a separate occupational status index for 1837–1939 because over the long interval 1837–2021 many new occupations emerged, and the social status of some occupations changed significantly. Thus the 1837–1939 index shows a much higher father-son correlation of 0.71 1837–1879, compared to only 0.58 for the CAMSIS 1990 index for that period. Fig 2, showing an absence of significant hypergamy 1837–1939, would be unchanged if we instead measured father status 1837–1939 using the CAMSIS 1990 index.

## Marriage and birth register data

**Marriages, 1912–2007.**    We compiled a database of all 30,769,942 registered marriages in England and Wales, 1912–2007. This was created by downloading the individual index entries from two websites: freebmd.org.uk (1912–1980) and familysearch.org (1980–2007). For marriages in this interval, the marriage index contains the full name of both bride and groom. By assigning status to rarer surnames we can test for hypergamy in marriage. We can also test whether women show more social mobility through marriage than men. Further we can estimate the relatively likelihood of men and women across the surname status distribution entering marriage. For each surname there should be a roughly equal stock of men and women in the prime marriage ages of 18–50. So we can measure relative marriage rates by social status just by comparing the total number of men and women at each status level who marry.

**Births, 1912–2007.**    We similarly compiled a database of all 67,670,339 births in England and Wales 1912–2007 using the sources as above. In these years the birth index always records the birth surname of the mother. If the child has a different surname to that of the mother, then that child surname will be the surname of the father. Thus for the great majority of births 1912–2007 we observe both parent surnames. Since in recent years many people cohabit without formal marriage, this gives an alternative measure of assortment in partnering, though one that depends on fertility. We can again test for hypergamy in such unions. We can also measure the relatively likelihood of men and women across the surname status distribution entering such unions.

**Surname status, 1999.**    We assign an average status to each surname in England and Wales using the Electoral Register of 1999. This 1999 register was a complete register of all voters, including detailed addresses. In the UK in 2018 85% of UK citizens eligible to vote were registered [38]. These addresses can be linked to the land registry to estimate average house values by postcode, for sales 1995–2005. Since there are 1,758,312 postcodes in the UK each postcode typically covers less than 20 houses. Average house value by postcode 1995–2005 (in 2017 prices) ranged from £8,000 to £24,000,000 with a pronounced right skew. We thus use the log housing value since house values have a close to log normal distribution. So our index of status for rarer surnames 1912–2007 is the average log housing value (2017 prices) associated with each surname 1995–2005.

Surname status on this measure varies much more for rarer surnames than for common ones. But for the rarest surnames random noise becomes an issue. For many surnames in the 1–10 frequency range in the electoral register the recorded surname will be just a transcription error. Also the house value information will be heavily affected by random noise.

For both marriages and births throughout the years 1912–2007, surname status correlates across bride and groom, as Fig 1 shows, as well as across mothers and fathers. As the figure

shows that correlation rises as surnames become rarer. Surnames contain more information about the average social status of its holders the rarer is the surname. In what follows we trade off information content of surnames and sample size by concentrating on surnames which appear 10–500 times in the 1999 electoral register. For marriages we find 1.8 million in England and Wales 1912–2007 where both parties had surnames in this size range. As Fig 1 shows the correlation between partners in surname status for surnames in this range was close to constant 1912–2007. We show in S1 File that postcode average house values in 1999 correlate well with occupational status for men, where we have data on both.

We will see below no significant female hypergamy in marriage or partnerships in England 1837–2021. But could there still be a difference in marital matching between men and women, of the form shown in Fig 6(A)? Could high status women be marrying down more than men, and low status women also marrying up more than men? That is, could we find a situation such as in Fig 6(A), where there was no difference in average status for men and women at marriage, but a different slope connecting their family status with that of their partner?

One situation we might think could create this would be if men sought both status and physical attractiveness in women in marriage, but women considered only status. Also assume physical attractiveness was equally distributed in women across the family status spectrum. While the average family status of husbands would still be the same as for wives, that would potentially lead women to experience more regression to the mean in marriage partner social status, and consequently more social mobility.

However, the intuition in the paragraph above is incorrect if men and women have equal probabilities of marriage across the status distribution. Even if men value physical appearance and women do not, this will create a status match between men and women that is symmetrical to that between women and men if all men and women marry. It is true that this will reduce the correlation in family status between marital partners, but with symmetrical effects on both sides of the marriage market.

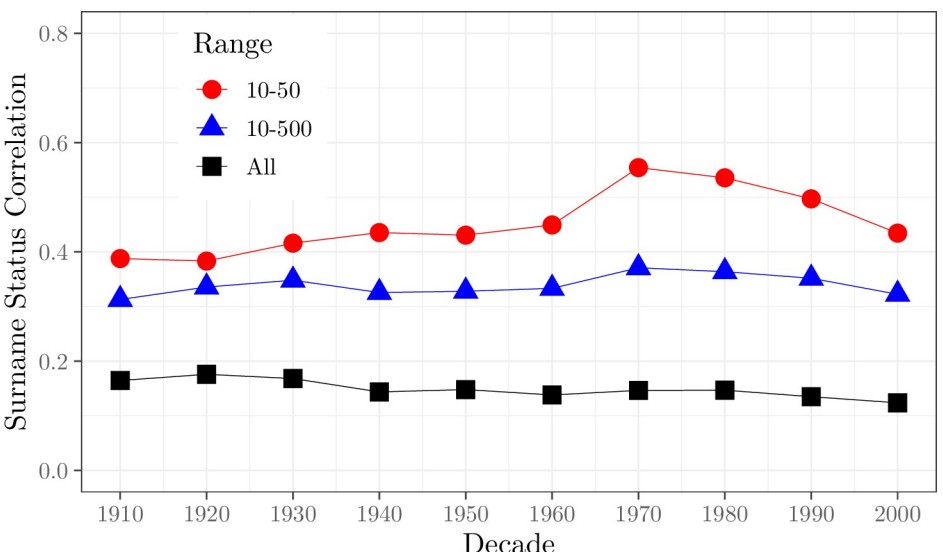

**Fig 1. Correlations of surname status in marriage, by marriage decade.** *Sources*: Marriage register, 1912–2007, Electoral Register, 1999, Land Registry 1995–2005. *Notes*: The figure shows the correlations between bride and groom average surname ln house value 1999, for three groups: all surnames, surnames held by 10–500 voters, and surnames held by 10–50 voters.

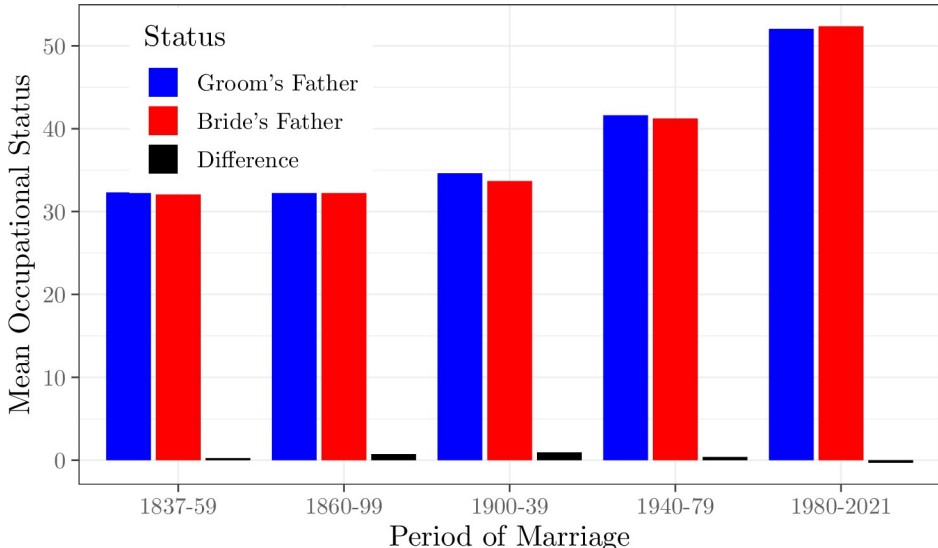

**Fig 2. Occupational status differences in marriages, 1837–2021.** *Notes*: Status is measured as the occupational status of the groom and bride's fathers, on a 0–100 scale. Hypergamy is measured as the difference in average father status between groom and bride.

To see this let $y_g$, $y_b$ be the social status of grooms and brides, assumed mean 0 and the same variance. For grooms $y_g$ alone measures their rank on the marriage market. However, assuming grooms value in brides also physical appearance, measured as a random shock u, mean 0, independent of bride social status. Brides' marital ranking is then

$$x_b = \varphi(y_b + u)$$

where $\varphi < 1$ is chosen such that $\sigma_{(y_b)}^2 = \sigma_{(x_b)}^2$. If we now match brides and grooms in marriage so that $y_g = x_b$, then we have $y_g = \varphi(y_b + u)$. In this case if we regress $y_g = by_b$, and alternately regress $y_b = cy_g$ then the expected value of $b$ will be the same as for $c$.

$$E(\hat{b}) = \varphi = E(\hat{c})$$

Despite the asymmetry in ranking between men and women, the only effect of men also ranking by physical appearance is to create both for men and women the same increase in regression to the mean in marital partners' social status. High social status males marry women on average lower in social status in search of more attractive partners. But high social status women also marry men on average lower in status, because if they are not physically attractive their best match is with a man lower in social status.

For there to be a difference in the matching across family status, there has to be a different partnering rate across the family status distribution for men and women. But that difference will be detected as a difference in average family status of women versus men.

## Results

### Average bride and groom family status, 1837–2021

For the 1.7 million parish marriage records we can measure average occupational status for the fathers of both brides and grooms. In this dataset there is no significant hypergamy by women in English marriage throughout 1837–2021. Overall matching is on average that of social equals, as measured by father occupational status for both bride and groom. This is shown in

Fig 2. As the figure shows grooms have a slight advantage in status. But while this is statistically significant it is quantitively insignificant. At its maximum for marriages 1900–39 the groom family status is one point higher than the bride family status, on a hundred-point scale. The average woman gained little in social status through marriage. By 1980–2021 brides were marrying grooms of on average lower family status than the brides, though again by inconsequential amounts.

As noted, while the church register data is largely representative of marriages as a whole for the years 1837–1914, thereafter it becomes a steadily smaller share of all marriages. So for the years 1912–2007 the general marriage index provides a more comprehensive measure of hypergamy.

Fig 3, panel (a) shows by decade the implied percent difference in average groom surname log house value minus average bride surname log house value for all registered marriages

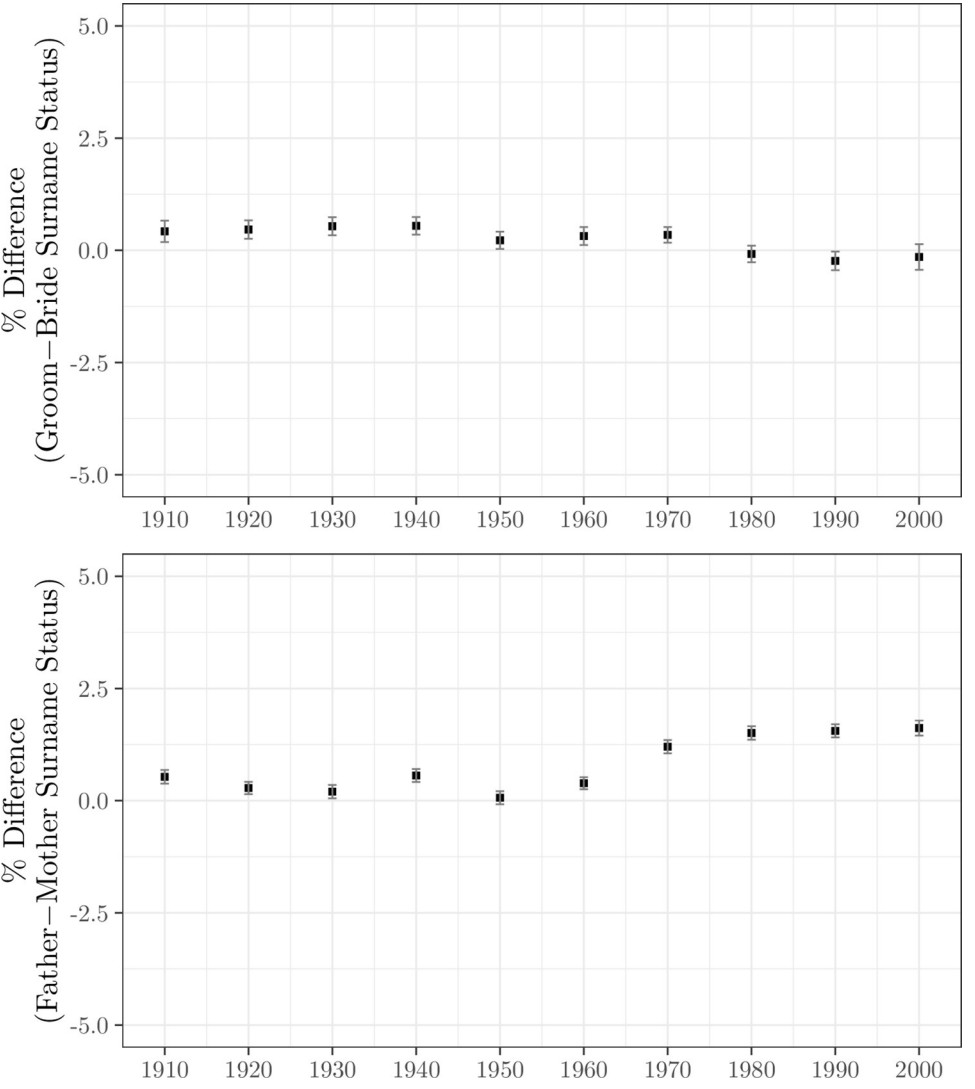

**Fig 3. Surname status differences, marital partners and parents, 1912–2007.** (a) Marriages, (b) Births. *Notes*: Status is measured by the average ln house value of surnames based on their addresses in the 1999 electoral register. Hypergamy is measured as the percent difference in average log house value for the groom's surname minus the bride's, for surnames held by 10–500 voters in 1999.

1912–2007, by decade. This is just the difference in average log house values for grooms versus brides, expressed as a percentage. As Fig 3(A) shows the difference in average log house value averages 0.2%. In the decades of the 1980s to 2000s it becomes slightly negative.

Since we have in the house value measure an attenuated measure of family status for each individual, an average across holders of their surname, the actual differences in status will be greater than estimated in Fig 3(A). However, Fig 1 shows that the degree of attenuation will be modest. For the estimated correlation between spouses in status is around 0.30 for names in the 10–500 range. Given the true individual correlation in spouse family status, measured by house value, will be 0.5 or less, this implies that the true difference in average status will be at maximum double the observed difference, using this measure. That implies differences in average house value for men relative to women will still typically be less than 0.4% for the families of marital partners.

A feature of modern partnerships, however, is that many couples never formally marry, even when they raise children together. By 2021 the majority of births for the first time were to mothers not married or in domestic partnerships. So an alternative measure of hypergamy comes from looking at the relative family status of fathers versus mothers. This measure, however, excludes childless couples. In about 1% of births the child surname is the same as the mother's. This can be because the father has the same surname as the mother, because the parents give the child the surname of the mother, or because a father is not listed on the birth certificate. Given this ambiguity we exclude such cases from the calculation of hypergamy.

Fig 3, panel (b) shows hypergamy by decade 1912–2007 measured at the level of parents. Again there is a modest tendency to female hypergamy, but with a measured average log house value difference of 0.8% or less between bride's family and groom's. If we correct for attenuation in the surname status measure these differences would be magnified, but still likely no more than 1.5%, and so still quantitatively insignificant.

The slightly greater measured hypergamy at the level of births compared to marriages would potentially be explained by lower surname status women having higher fertility. Women at the bottom of the surname distribution will typically have partners of higher social status. If such women are overrepresented in the birth data we would get an appearance of hypergamy even though at the partnership level there may be no such effect. This explanation requires, however, that fathers do not show the same strength of decline in fertility with surname status. Otherwise these fertility effects would cancel out with no appearance of hypergamy.

Church marriage records 1837–2021 record for most brides and grooms their age at marriage. For women, age can be taken as a correlate of physical appearance, with younger women more physically attractive in the marriage market. Supporting this, a study looked at speed dating, where men and women with no prior information about each other, conversed for 3 minutes, before choosing whether they desired further contact. The major factors determining positive choices were all physical, for both women and men. Women with lower BMI, younger age, and greater self-rated facial attractiveness generated more positive responses. Women's height had no effect. For women age and positive partner responses had a highly statistically significant negative correlation [7]. Similarly a study of online dating behavior showed that, controlling for social status, there was a strong revealed preference by men for younger women. In contrast women had a weak positive preference for older men [8]. Consistent with this, male ratings of the attractiveness of female faces declined more with age than did female ratings of the attractiveness of male faces [9].

We can then test with these church marriage records whether there was any connection between the age gap between marriage partners, and the difference in occupational status of their fathers. If female hypergamy was driven by men trading status for physical appearance,

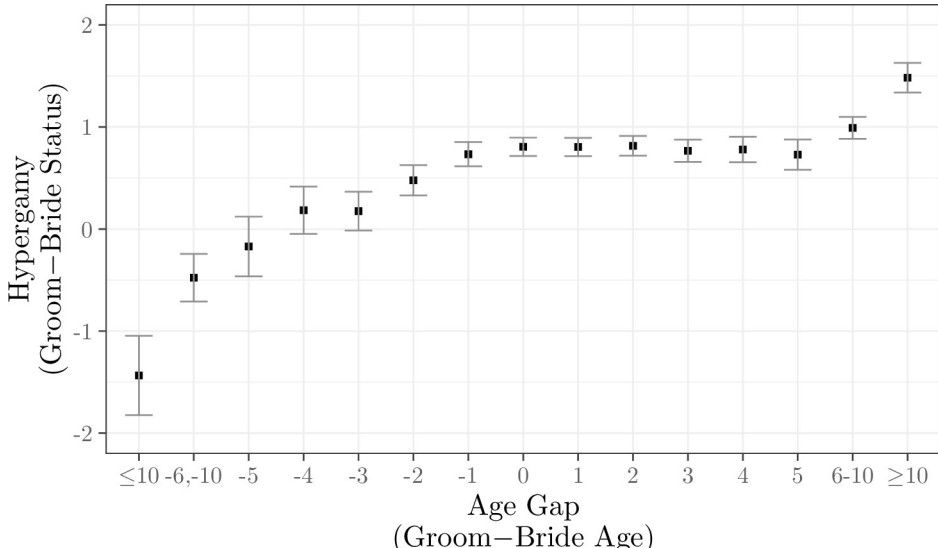

**Fig 4. Age difference at marriage and female hypergamy, father occupational status, 1837–2021.** *Notes*: The hypergamy measure is the difference between bride and groom father occupational status, on a scale of 0–100. The age gap is integer values of groom age in years minus bride age. The bars indicate 95% confidence intervals.

then the larger the age gap, the greater the expected difference in family status between bride and groom. Fig 4 shows the difference between groom and bride father status as a function of the age difference in marriage 1837–2021.

There is a positive association between the age gap and the father status gap. But as we see in Fig 4, the average difference of about 0.5 between groom father and bride father on a scale of 0–100 is quantitatively insignificant. Though the status gap is higher for age gaps of 10 or more years in both directions, it is still modest even for these extreme age gaps, at around 1.5 points on the 100 point scale.

## Marriage rates by social class, men versus women

Consistent with the evidence of equivalent family status on average for men and women in marriage, and in parenting, in England 1837–2021, there is no evidence of gender differences in marriage rates by family status.

From the parish register data, relative marriage rates were the same for men and women across the social spectrum, as measured by their fathers' occupational status. This is shown in Fig 5, panel (a) for church marriages 1837–59. The figure shows the share of all marriages by men and women with a given father status. There may be differences in the propensity to marry by father status, but Fig 5(A) shows that the relative propensity to marry for men and women was equal across the father status distribution.

The same constant relative marriage rates are also shown in Fig 5, panel (b) for recent church marriages, 1980–2021. Note that Fig 5 does not exclude the possibility that there were different propensities towards multiple marriages across men and women, and across social classes. It just establishes that counting at the level of marriages, equal numbers of men and women from each social class were represented.

The general marriage register shows a similar equal marriage rate for men and women across the surname status distribution throughout 1912–2007. This is illustrated in Fig 5, panel (c) using rarer surnames, those with frequency 10–500 in the 1999 electoral register, for marriages 1980–2007.

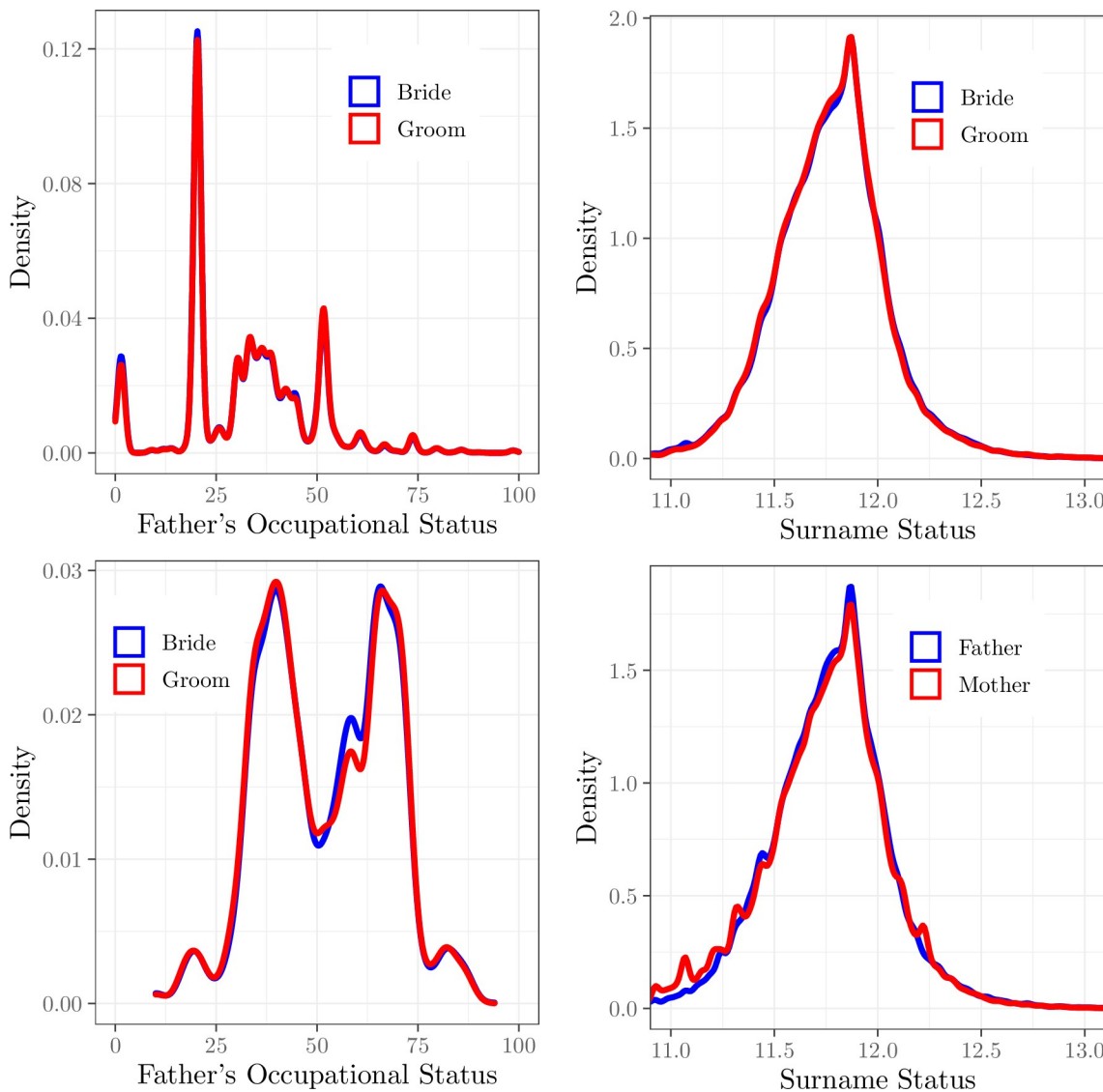

**Fig 5. The distribution of father occupation status, and surname status, at marriage, and parentage, for men and women, England, 1837–2021.** (a) Occupation, 1837–99, (b) Occupation, 1980–2021, (c) Surname Status, Marriages, 1980–2007, (d) Surname Status, Births, 1980–2007.

However, when we turn to births 1980–2007 in Fig 5, panel (d) we see for the bottom decile of surname status an excess of women. A possible explanation of this is not that low status women are able to "partner up" differentially, but that they have higher fertility than men of low status. This is speculation only, however. Note also that this excess of women in the lowest decile of family status for births translates into a less than 1% higher average house value for men as opposed to women as parents for children born 1980–2007. So overall there is even here weak sign of female hypergamy.

## The nature of marital sorting

Above we see that men and women on average matched in marriage to partners of equivalent birth-family status. But that still potentially leaves a possibility, portrayed in Fig 6, panel (a)

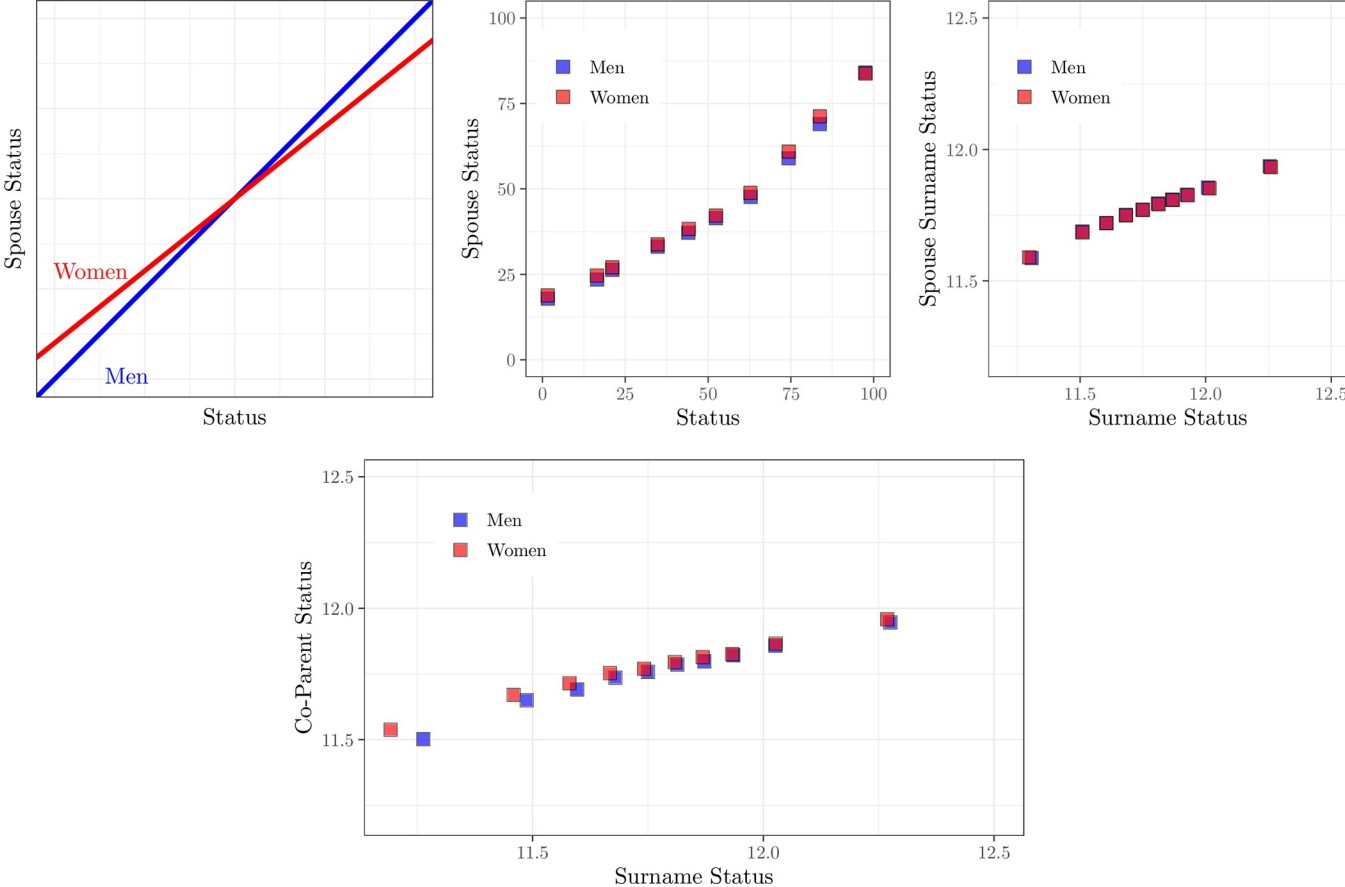

**Fig 6. Spouse status relative to own status, marriages and births, 1837–2021.** (a) Hypothetical, (b) Occupational Status, Marriages, 1837–99, (c) Surname Status, Marriages, 1980–2007, (d) Surname Status, Births, 1980–2007. *Notes*: Panels (b)-(d) split male and female status into deciles, and then calculate the average status score of partners in that decile.

where men and women match differently in marriage across the status spectrum. In Fig 6(A), high status women match with relatively lower status men than do high status men with women. And low status women match with relatively higher status men compared to low status men.

However, with both the church register marriage data, or the larger national register data, we find no evidence of greater regression to the mean in marriage by women as compared to men, and hence greater social mobility of women through marriage.

If we plot for men and women the status of their spouse's father relative to that of their own father, we see mostly the same relationship across the status distribution for women as for men. This is shown in Fig 6, panel (b), for church marriages 1837–59. Women with fathers of status rank 97, for example, marry men whose fathers have an average status rank of 83. But the same is true for men with fathers of this rank, and the average status of their wives' fathers.

For the most recent years, marriages or births 1980–2007, Fig 6(C) shows similarly by decile of surname status, for surnames of frequency 10–500 in the 1999 electoral register, the average partner surname status as a function of men's or women's surname status (by deciles). As can be seen, for marriages the slopes for men and women are the same.

For births 1980–2007 the bottom decile of women has modestly lower status than the bottom decile of men, as shown in panel 6(d). For this bottom decile there is modest indication of

"marrying up" relative to men in the bottom decile. As discussed above this effect could be produced if women in the bottom decile had higher fertility than men in this decile.

## Discussion

The absence of evidence of hypergamy in England and the constancy across the status spectrum of male and female marriage rates in the years 1980–2021 in England is surprising in light of the recently published paper on hypergamy in Norway [11]. The authors of this paper note in their abstract *"hypergamy is an important feature of today's mating patterns."* And the Norwegian paper, like this one, uses family background to measure status.

The mechanism driving hypergamy in modern Norway is that high family status men have a greater relative probability than high family status women of ever partnering, as well as a greater chance of partnered more than once. However, despite the abstract of the paper, it should be noted that the overall extent of hypergamy in Norway is very limited. The average status of all men and women in the study, measured as their parents' income percentile, was equal at 50.6%. The average status of those ever partnering was 51.6% for the family income of men, and 51.2% for women, a difference of 0.4% on a scale 0–100% (Almås et al., 2023, page 9). The Norwegian data is thus completely consistent with the English data on birth parent status for births 1980–2007, where there also are signs of very modest hypergamy.

As noted above, we consistently measure hypergamy relative to birth-family status of husband and wife. If daughters' attained status systematically declined relative to birth-family social status, in a way that sons' status did not, then there would be a form of hypergamy. But we find no evidence that brides in England 1837–2021 were regarded as having degraded status relative to their birth families.

If men prize physical attributes in mates which are uncorrelated with family status, then the correlation of social status in marriage will decline. However there is ancillary evidence that the importance of physical attributes in forming matches must always have been modest. In a related paper we estimate the correlation of underlying social abilities for brides and grooms in marriage in England as constant at around 0.8 1837–2021 [39].

This very strong marital correlation implies that either the importance for men in making a match based on physical appearance was limited, or it must be that physical appearance in women was strongly correlated with social abilities. For if physical appearance was given the same weight in matching as social abilities, and was uncorrelated with social abilities, the correlation of marital partners with respect to social abilities would be at maximum 0.5. If physical appearance was uncorrelated with social status, to produce a marital correlation of 0.8 in social status, the weight given to physical attributes in marriage would be at maximum one quarter of that given to social abilities.

## Supporting information

**S1 File. This is a set of supplementary tables and figures.**
(PDF)

## Acknowledgments

We thank the members of the volunteer organizations, FreeReg and FreeBMD, for their efforts transcribing and making freely available on the web transcriptions of the Civil Registration index and of parish marriage records. Without their contributions this paper was not possible. We thank also Ziming Zhu of LSE for excellent research assistance.

## Author Contributions

**Conceptualization:** Gregory Clark, Neil Cummins.

**Data curation:** Gregory Clark, Neil Cummins.

**Formal analysis:** Gregory Clark, Neil Cummins.

**Investigation:** Gregory Clark, Neil Cummins.

**Methodology:** Gregory Clark, Neil Cummins.

**Project administration:** Gregory Clark, Neil Cummins.

**Resources:** Gregory Clark, Neil Cummins.

**Supervision:** Gregory Clark, Neil Cummins.

**Validation:** Gregory Clark, Neil Cummins.

**Visualization:** Gregory Clark, Neil Cummins.

**Writing – original draft:** Gregory Clark, Neil Cummins.

**Writing – review & editing:** Gregory Clark, Neil Cummins.

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
