## [Decision Letter · Decision Letter 0]

4 Jul 2024

PONE-D-24-22853Hypergamy Reconsidered: Marriage in England, 1837-2021PLOS ONE

Dear Dr. Cummins,

Thank you for submitting your manuscript to PLOS ONE. After careful consideration, we feel that it has merit but does not fully meet PLOS ONE’s publication criteria as it currently stands. Therefore, **we invite you to submit a revised version of the manuscript that addresses the points raised during the review process.**

Thank you for your submission to Plos One. The referees find the manuscript interesting and appropriate for the journal, but also feel that substantial changes must be made before it can be considered publishable. I would be glad to consider a revised version of your manuscript that takes these comments and suggestions into account. 

Please submit your revised manuscript Aug 18 2024 11:59PM**.** If you will need more time than this to complete your revisions, please reply to this message or contact the journal office at plosone@plos.org. Please include the following items when submitting your revised manuscript:A rebuttal letter that responds to each point raised by the academic editor and reviewer(s). You should upload this letter as a separate file labeled 'Response to Reviewers'.A marked-up copy of your manuscript that highlights changes made to the original version. You should upload this as a separate file labeled 'Revised Manuscript with Track Changes'.An unmarked version of your revised paper without tracked changes. You should upload this as a separate file labeled 'Manuscript'.

We look forward to receiving your revised manuscript.

Kind regards,

Grażyna Liczbińska

Academic Editor

PLOS ONE

Journal Requirements:

3. Please ensure that you include a title page within your main document. You should list all authors and all affiliations as per our author instructions and clearly indicate the corresponding author.

Reviewers' comments:

Reviewer's Responses to Questions

**Comments to the Author**

1. Is the manuscript technically sound, and do the data support the conclusions?

Reviewer #1: Partly

Reviewer #2: Yes

2. Has the statistical analysis been performed appropriately and rigorously? 

Reviewer #1: No

Reviewer #2: Yes

3. Have the authors made all data underlying the findings in their manuscript fully available?

Reviewer #1: No

Reviewer #2: No

4. Is the manuscript presented in an intelligible fashion and written in standard English?

Reviewer #1: Yes

Reviewer #2: Yes

5. Review Comments to the Author

Reviewer #1: The paper deals with an interesting topic, which is presented in a long time perspective. It is regrettable that it contains some problematic structural and methodological elements. The text's structure is confusing and reader-unfriendly, as the part concerning on data analysis and methodology is presented only at the end, after the results and discussion. This reduces the possibility of understanding the text. The data in question was not accessible at the time of the review of the paper. The authors have stated that they will publish it when the paper is published.

With regard to the content, the first problem lies paradoxically in what should be the article's strength, namely its long time span. The years 1837-2021 represent a period of significant social transformation, including changes in the way people choose a partner. In this respect, the authors discuss only two factors: the social (comparing the occupations of the fathers of the groom and the bride) and the physical appearance of women, which was used as a counterbalance to the lack of social background. It seems likely that there were a great many other considerations involved in mate selection, particularly after 1914 (and perhaps even more so after 1990). Nevertheless, it is appropriate to evaluate the hypergamy theory for the population entering after 1914, specifically to ascertain whether there is a disparity in the social status of grooms and brides. However, a significant challenge arises when considering the categories being compared. If the objective was to test the hypothesis that women married men who were older and of higher social status than their own families, it is unclear why the grooms' own status was not taken into account. In order to ﬁnd out whether women actually adopted a strategy whereby they wanted to achieve social mobility through marriage, it is necessary to examine the social status of their grooms, rather than that of the fathers of the grooms. An interesting and appropriate analysis would be to ascertain whether the status of the grooms and their fathers differed. The second contentious aspect is that the fact that a woman marries a man who does not have a higher social status than her at the time of marriage does not preclude the above strategy. This is because many social positions also implied the potential for significant social mobility in the future. To illustrate, if a woman from an artisanal family married a law student or a recent graduate who was, for example, working as a lower law clerk at the time of marriage (which menas that his present social status was similarly of a low rank), and this groom also came from an artisanal family, it can be observed that although formally both fiancés will come from the same social background, the hypergamy theory may still be applicable. This is because the woman may have envisaged in her marriage strategy that her fiancé would attain the position of a lawyer, which would represent a significant social shift.

Consequently, it is not possible to apply the same methodology to the post-1914 period as to the earlier period. One reason for this is that women were no longer deriving their social status from their fathers. Therefore, as in the case of grooms, one should work with the social status of brides, not just their fathers. The second, more serious problem is the source base. The authors acknowledge that their data is primarily based on church registers. They also state that in 1914 the proportion of civil marriages was already 24% and rising (reaching 68% in 2012), and they admit that this fact may introduce bias into their results. In addition, the authors also work with an analysis that relates social status to surname for this period. Two potential risks must be considered. Firstly, the fact that someone lives at a particular address does not necessarily mean they are a homeowner; they may just live in a rental property, which generates a completely different property status. Secondly, the social distribution obtained from the occurrence of surnames in 1999 cannot be mechanically applied to the whole period 1912-2007. This is because it is not only social mobility but also geographical mobility that must be considered.Firstly, the fact that someone lives at a particular address does not necessarily mean they are a homeowner; they may just live in a rental property, which generates a completely different property status. Secondly, it is unclear whether the social distribution obtained from the occurrence of surnames in 1999 can be mechanically applied to the whole period 1912-2007. This is because the phenomenon of social mobility must be considered in conjunction with that of geographical mobility.

For the aforementioned reasons, I would recommend that the authors limit their study to the period 1837-1914 and elaborate their analysis in greater depth (by adding a comparison of the social status of grooms and their fathers-in-law). Furthermore, I would suggest that they consider whether a coefficient expressing the potential for social growth could be assigned to the various occupations held by men at the time of marriage. It is not the occurrence of social mobility that is important for confirming the hypergamy theory; rather, it is whether it could have occurred. This factor must be considered in the context of the marital strategy. Furthermore, the age of the fiancés should be included in the analysis. Finally, the analysis should be structured more by social groups to avoid the overall result masking the differences between groups. It is conceivable that a specific social group may appear to have a higher propensity for hypergamy than another, but if that group is not large enough, its influence on the overall results will be negligible. Additionally, it is crucial to consider the potential impact of the overall social structure of the society on the results. If a society is relatively poorly differentiated in the sense that the majority of its members belong to the lower (agricultural) strata and, for example, people with a university education represent at most a unit of one percent, then it is clear that social shift is difficult. In such a case, it would be necessary to create a suitable scale that captures even a less pronounced social shift. Related to this is the determination of social status itself, which I believe is inadequately explained in the article. The authors merely state that they assigned occupational scores to fathers using the CAMSIS 1990 scales of social status and that they then constructed their own occupational status for marriages 1837-1939 (as they did for marriages 1940-2021?). The authors do not explain why they did not use another widely applied system, HISCO/HISCLASS. If one is working with a custom social scale, it is important to specify it in more detail. This is because the question is whether the presented conclusion that hypergamy was negligible is influenced by the fact that any social differences were masked by the coding used.

In conclusion, I believe that this paper has the potential to yield significant results once the analysis is deepened. These results will make a valuable contribution not only to hypergamy theory, but also to social mobility research in general.

Reviewer #2: The paper provides a thoughtful consideration and clear answer to the proposed research question. It also deserves credit for making the still rare distinction between statistical and quantitative significance (especially when the sample size is large). Some minor issues need to be considered before publication to make the paper more accessible to a general audience:

1) If possible, please move the materials and methods section to the usual place between introduction and results (then it would be unnecessary to explain 10-50 and 10-500 frequencies if they appear in the text).

2) Does the study assume relative stability of surname status in England during the 19th and 20th centuries (it seems so, as historical records are linked to 1999 house values)? Are there any studies that confirm or refute this claim? Please cite and briefly discuss them, even if they are self-citations.

3) Please make the list of sources clear. In total there are three types of sources:

1) Database of 1.7 million church marriages in England 1837-2021

2) Population register data on marriages and births in England by surname, 1912-2007

3) House value by surname in 1999

Is this correct?

4) You write "Despite the widespread assumption in the anthropology and sociology literatures that hypergamy of this type was and is a common social practice" - Please cite the sources to support this claim (even if they are referred to in the introduction).

5) Please explain the 10-500 and 10-50 range, it appears for the first time on page 4 with no explanation (only if the materials and methods section remains at the end).

6) Figure 1b - could you reduce the scale of the y-axis - the changes in differences and CIs are barely visible.

7) You write - "The slightly greater measured hypergamy at the level of births compared to marriages would potentially be explained by women with lower surname status having higher fertility". - Is this an actual inference from the data you have or is it based on the literature? Please clarify and cite some appropriate numbers.

8) Technicalities:

8) Typographical errors:

a) Typos: bridge instead of bride - p. 7, p. 14

b) All tables and some figures (e.g. S1) are missing.

c) Please check the numbers of the figures, as they do not seem to correspond correctly to the content (e.g. Figure 1 is quoted on page 14 and it should be figure 4)

6. PLOS authors have the option to publish the peer review history of their article (what does this mean?). If published, this will include your full peer review and any attached files.

Reviewer #1: **Yes: **Alice Velková

Reviewer #2: No

---

## [Author Response · Author response to Decision Letter 0]

19 Aug 2024

Dear Grażyna

Below please find our point by point response to the two reviewers (highlighted in yellow). Both reviewers called for a reorganization of the order of presentation in the paper, and we have made this change presenting all the “Methods and Materials” content before the empirical results.

Both reviewers called for some demonstration that surname status measured in 1999 would also apply to 1912, and we have supplied that.

We were able to respond to all the other suggestions of Referee #2, which were many calling for more documentation of claims and clarifying changes.

Referee #1 made a large number of additional suggestions for revision, many of which would have fundamentally changed the paper – such as restricting the analysis to 1837-1914, or recasting the paper in terms of a comparison of grooms to father-in-law. 

Here we have added a figure to respond to their suggestion about explicitly taking into account marriage age. 

Otherwise we have mainly tried to explain, sometimes with additional data and graphics, why we have not adopted these suggestions. 

But we have added to the Supplementary Materials file some of the material produced in response to these referee suggestions as illuminating these issues with measuring hypergamy.

Once the content of the paper is finalized we will post replication files to Harvard Dataverse. We have not done this in advance of acceptance in case there will be further revisions in the content of the paper.

Response to Reviewers

Reviewer #2: The paper provides a thoughtful consideration and clear answer to the proposed research question. It also deserves credit for making the still rare distinction between statistical and quantitative significance (especially when the sample size is large). Some minor issues need to be considered before publication to make the paper more accessible to a general audience:

1) If possible, please move the materials and methods section to the usual place between introduction and results (then it would be unnecessary to explain 10-50 and 10-500 frequencies if they appear in the text).

This rearrangement has been done. All discussion of materials and methods now occurs before the discussion of estimation results.

2) Does the study assume relative stability of surname status in England during the 19th and 20th centuries (it seems so, as historical records are linked to 1999 house values)? Are there any studies that confirm or refute this claim? Please cite and briefly discuss them, even if they are self-citations.

In the supplementary materials we show an alternative way of attributing surname status, which is based on wealth at death. With this method we estimate surname status separately for 1910-39, 1940-79, and 1980-92. In table S8 we show these surnames statuses correlate significantly over the three intervals. 

3) Please make the list of sources clear. In total there are three types of sources:

1) Database of 1.7 million church marriages in England 1837-2021

2) Population register data on all marriages and births in England by surname, 1912-2007

3) House value by surname in 1999

Is this correct?

Yes. All three sources are described in more detail in the Supplementary Materials file, which will be available online. 

4) You write "Despite the widespread assumption in the anthropology and sociology literatures that hypergamy of this type was and is a common social practice" - Please cite the sources to support this claim (even if they are referred to in the introduction).

Sources now cited and included in reference list.

5) Please explain the 10-500 and 10-50 range, it appears for the first time on page 4 with no explanation (only if the materials and methods section remains at the end).

When we look at status using surnames there is a tradeoff in surname frequency in the 1999 electoral register between surname count, sample sizes, and the ratio of status signals to noise. Less frequent surnames vary more in the average house value by surname than do common surnames. But very rare surnames contain many that are just transcription errors, and where the house value information on surname status is just noise. This argues for setting a lower limit on surname frequency in the 1999 register for a surname to be included. As figure 1 shows in terms of the correlations in status of rare surnames across marital partners this would argue for using surnames in the 10-50 frequency range in 1999. But widening the rare surname frequency range to 10-500 greatly increases the sample size and the precision of the estimates of hypergamy.

6) Figure 1b - could you reduce the scale of the y-axis - the changes in differences and CIs are barely visible.

Figure 1b is now figure 3a. The vertical scale has been reduced to show better the estimates and confidence intervals.

7) You write - "The slightly greater measured hypergamy at the level of births compared to marriages would potentially be explained by women with lower surname status having higher fertility". - Is this an actual inference from the data you have or is it based on the literature? Please clarify and cite some appropriate numbers.

This is purely speculation on our part. We have clarified this in the text.

8) Technicalities:

a) Typos: bridge instead of bride - p. 7, p. 14

b) All tables and some figures (e.g. S1) are missing.

c) Please check the numbers of the figures, as they do not seem to correspond correctly to the content (e.g. Figure 1 is quoted on page 14 and it should be figure 4)

Thanks. Corrected

Reviewer #1: The paper deals with an interesting topic, which is presented in a long time perspective. It is regrettable that it contains some problematic structural and methodological elements. The text's structure is confusing and reader-unfriendly, as the part concerning on data analysis and methodology is presented only at the end, after the results and discussion. This reduces the possibility of understanding the text. The data in question was not accessible at the time of the review of the paper. The authors have stated that they will publish it when the paper is published.

Both reviewers raise the issue of the structure of the paper, and ask for all the “materials and methods” section to precede the results. We have thus restructured the paper substantially to follow this format.

Once we finalize the content of the paper, we will make available in the Harvard Dataverse all the data necessary to reproduce all figures and results in the paper.

With regard to the content, the first problem lies paradoxically in what should be the article's strength, namely its long time span. The years 1837-2021 represent a period of significant social transformation, including changes in the way people choose a partner. In this respect, the authors discuss only two factors: the social (comparing the occupations of the fathers of the groom and the bride) and the physical appearance of women, which was used as a counterbalance to the lack of social background. It seems likely that there were a great many other considerations involved in mate selection, particularly after 1914 (and perhaps even more so after 1990). Nevertheless, it is appropriate to evaluate the hypergamy theory for the population entering after 1914, specifically to ascertain whether there is a disparity in the social status of grooms and brides. However, a significant challenge arises when considering the categories being compared. If the objective was to test the hypothesis that women married men who were older and of higher social status than their own families, it is unclear why the grooms' own status was not taken into account. In order to ﬁnd out whether women actually adopted a strategy whereby they wanted to achieve social mobility through marriage, it is necessary to examine the social status of their grooms, rather than that of the fathers of the grooms. An interesting and appropriate analysis would be to ascertain whether the status of the grooms and their fathers differed. 

We use father occupational status, or surname status, as a measure of the family status of men and women entering marriage. To compare the family status of bride and groom in marriage and parenting we need a symmetrical measure of family status, which both of these supply. 

Referee #1 calls for us to instead compare groom status with father-in-law status. Their reasoning on this is not entirely clear from their comments, but the concern seems to be that sons who married might have had significantly higher status than their fathers, and thus also than their father-in-laws, allowing for possible hypergamy.

However, when we compare average groom, father and father-in-law occupational status for marriages 1880-99 (287,000 marriages) we find for a range of 0-100 an average status if grooms – 33.5, fathers – 33.6, and father-in-laws 32.6. The results with grooms look just like those with fathers (see figure 1). This is why we have persisted with the basic organization of the paper, measuring family status of bride and groom.

The second contentious aspect is that the fact that a woman marries a man who does not have a higher social status than her at the time of marriage does not preclude the above strategy. This is because many social positions also implied the potential for significant social mobility in the future. To illustrate, if a woman from an artisanal family married a law student or a recent graduate who was, for example, working as a lower law clerk at the time of marriage (which menas that his present social status was similarly of a low rank), and this groom also came from an artisanal family, it can be observed that although formally both fiancés will come from the same social background, the hypergamy theory may still be applicable. This is because the woman may have envisaged in her marriage strategy that her fiancé would attain the position of a lawyer, which would represent a significant social shift.

Perhaps hypergamy exists because women marry men who at the time of marriage have no higher status than their fathers, but who will over the course of their careers gain much greater occupational status. Using the church marriage data we do see for marriages 1837-69 that older grooms have on average higher status, as is shown in the figure below (dotted line). However, also in the figure we also plot occupational status of grooms by age controlling for father occupational status. Now the rise in occupational status of grooms with age is seen to mainly come from grooms from lower status families marrying earlier. Controlling for father status there is little rise in groom status with age. Groom status at marriage is close to expected groom status at age 40 or later. There is no evidence for the suggested mechanism for hypergamy.

We have added this discussion, and the figure, to the Supplementary Materials.

Groom occupational status 1837-69 by age, and controlling for father occupational status

Consequently, it is not possible to apply the same methodology to the post-1914 period as to the earlier period. One reason for this is that women were no longer deriving their social status from their fathers. Therefore, as in the case of grooms, one should work with the social status of brides, not just their fathers. 

In this paper we seek to measure hypergamy consistently across time in terms of the status of the families women and men grew up in. This allows us to examine the entire period 1837-2021.

The second, more serious problem is the source base. The authors acknowledge that their data is primarily based on church registers. They also state that in 1914 the proportion of civil marriages was already 24% and rising (reaching 68% in 2012), and they admit that this fact may introduce bias into their results. 

We acknowledge in the paper the increasing selectivity of the source post 1914. But significant numbers of people still were married in church post 1914, and for them it is interesting to ask what the extent of hypergamy was. We are able to supplement the church record with general marriage records post 1912. 

In addition, the authors also work with an analysis that relates social status to surname for this period. Two potential risks must be considered. Firstly, the fact that someone lives at a particular address does not necessarily mean they are a homeowner; they may just live in a rental property, which generates a completely different property status. Secondly, the social distribution obtained from the occurrence of surnames in 1999 cannot be mechanically applied to the whole period 1912-2007. This is because it is not only social mobility but also geographical mobility that must be considered. Firstly, the fact that someone lives at a particular address does not necessarily mean they are a homeowner; they may just live in a rental property, which generates a completely different property status. Secondly, it is unclear whether the social distribution obtained from the occurrence of surnames in 1999 can be mechanically applied to the whole period 1912-2007. This is because the phenomenon of social mobility must be considered in conjunction with that of geographical mobility.

Using an ancillary source we can plot male occupational status versus their postcode average house value 1999-2024. Housing value is a strong predictor of occupational status, as the figure below shows. Like any indicator of social status house value embodies errors. But for testing whether average family status of grooms equals that of brides, such random errors are not a problem. We have included the figure below in the Supplementary Materials.

As to the question of of whether the surname status estimates of 1999 also apply to 1912, note that in figure 1 the correlation of surname status between brides and grooms is just as strong as in 2007. In the supplementary materials we also estimate surname status by wealth at death 1910-39, 1940-79, and 1980-92. These measures are significantly correlated across time periods. Further on these period specific surname status measures we find in figure S3 very similar results to those using the surname status measures estimated from 1999. 

For the aforementioned reasons, I would recommend that the authors limit their study to the period 1837-1914 and elaborate their analysis in greater depth (by adding a comparison of the social status of grooms and their fathers-in-law). 

For the reasons stated above we have continued to analyze the whole interval 1837-2021. We also discuss above why the groom to father-in-law comparison would lead to the same conclusions as the father to father-in-law comparisons.

Furthermore, I would suggest that they consider whether a coefficient expressing the potential for social growth could be assigned to the various occupations held by men at the time of marriage. It is not the occurrence of social mobility that is important for confirming the hypergamy theory; rather, it is whether it could have occurred. This factor must be considered in the context of the marital strategy. 

As we discuss above groom occupational status at time of marriage (typically age 20-29) changes little over the life course.

Furthermore, the age of the fiancés should be included in the analysis. 

We have done an analysis now in the paper looking at father occupational status differences and the age gap between grooms and brides 1837-2021. This is reported in Figure 4: Age Difference at Marriage and Female Hypergamy, Father Occupational Status, 1837-2021

Finally, the analysis should be structured more by social groups to avoid the overall result masking the differences between groups. It is conceivable that a specific social group may appear to have a higher propensity for hypergamy than another, but if that group is not large enough, its influence on the overall results will be negligible. Additionally, it is crucial to consider the potential impact of the overall social structure of the society on the results. If a society is relatively poorly differentiated in the sense that the majority of its members belong to the lower (agricultural) strata and, for example, people with a university education represent at most a unit of one percent, then it is clear that

---

## [Decision Letter · Decision Letter 1]

30 Oct 2024

PONE-D-24-22853R1Hypergamy Reconsidered: Marriage in England, 1837-2021

PLOS ONE

Dear Dr. Cummins,

Thank you for submitting your manuscript to PLOS ONE. After careful consideration, we feel that it has merit but does not fully meet PLOS ONE’s publication criteria as it currently stands.  **While the reviewers have accepted your revision, you are additionally required to address certain points raised by the editorial office. **These comments have been the subject of discussion between myself and the editorial team. Therefore, we invite you to submit a revised version of the manuscript that addresses the points raised during the review process. 

We look forward to receiving your revised manuscript.

Kind regards,

Grażyna Liczbińska

Academic Editor

PLOS ONE

Journal Requirements:

**Comments from the editorial office:**

- The findings are largely dependent on the claim that the factor of age from the data is a proxy for physical attractiveness, the source of which is not referenced by the literature nor made clear. This claim appears throughout the manuscript, for example: "For women, age can be taken as a correlate of physical appearance, with younger women more physically attractive in the marriage market", which is not supported by any academic references. We feel that this claim should be made much more explicit in the Introduction (and throughout).

- In the Methodology, you use CAMSIS 1990 scores of social status for marriages 1940-2021. For marriage 1837-1939 you have constructed their own occupational status association index. While you have responded partly to reviewers query about this, we don't feel the submission fully explains why two different indexes were used and why the cut-off of 1939/1940 was chosen.

- Reviewer 1 commented on how fathers' social status may not be the same as their children's social status, and further exploration of parental status vs child status vs married status would be helpful, and while your response to this was considered, we feel this limitation should be at least explicitly stated.

- Additional consideration and discussion of the role of maternal status would be helpful

- Finally, the structure of the submission also remains unclear, and while the authors have made some changes related to this, we kindly request a clearer adherence to the PLOS ONE manuscript guidelines (https://journals.plos.org/plosone/s/submission-guidelines#loc-manuscript-organization).

Reviewers' comments:

Reviewer's Responses to Questions

**Comments to the Author**

1. If the authors have adequately addressed your comments raised in a previous round of review and you feel that this manuscript is now acceptable for publication, you may indicate that here to bypass the “Comments to the Author” section, enter your conflict of interest statement in the “Confidential to Editor” section, and submit your "Accept" recommendation.

Reviewer #1: All comments have been addressed

Reviewer #2: All comments have been addressed

2. Is the manuscript technically sound, and do the data support the conclusions?

Reviewer #1: Yes

Reviewer #2: Yes

3. Has the statistical analysis been performed appropriately and rigorously? 

Reviewer #1: Yes

Reviewer #2: Yes

4. Have the authors made all data underlying the findings in their manuscript fully available?

Reviewer #1: Yes

Reviewer #2: No

5. Is the manuscript presented in an intelligible fashion and written in standard English?

Reviewer #1: Yes

Reviewer #2: Yes

6. Review Comments to the Author

Reviewer #1: The authors have implemented proposed alterations to the structure and more effectively articulated the sources utilized, thereby enhancing the article's readability. The incorporation and enhancement of the results and arguments in the supplement are highly beneficial. Furthermore, I appreciate that the authors have responded to my previous comments in great detail and have made efforts to react to them. Despite some remaining reservations expressed earlier, I believe the paper's publication should not be impeded. Given the diversity of approaches among researchers and the influence of their respective specialisms, I welcome the publication of an article that may facilitate constructive debate on this topic.

Reviewer #2: (No Response)

7. PLOS authors have the option to publish the peer review history of their article (what does this mean?). If published, this will include your full peer review and any attached files.

Reviewer #1: **Yes: **Alice Velková

Reviewer #2: No

---

## [Author Response · Author response to Decision Letter 1]

25 Nov 2024

Responses to Editor Comments/Suggestions

- The findings are largely dependent on the claim that the factor of age from the data is a proxy for physical attractiveness, the source of which is not referenced by the literature nor made clear. This claim appears throughout the manuscript, for example: "For women, age can be taken as a correlate of physical appearance, with younger women more physically attractive in the marriage market", which is not supported by any academic references. We feel that this claim should be made much more explicit in the Introduction (and throughout).

our reply

Two responses on this.

(1) Only 1 out of 5 figures which present the main results of the paper depend on using age as a correlate of female physical attractiveness. The other 4 figures show the absence of female hypergamy in England 1837-2021 without any reliance on age being a proxy for female physical attractiveness.

(2) We now include in the text a brief summary of 3 recent scholarly papers which show that female age is indeed a correlate of physical attractiveness in the settings of speed dating and online dating. This is:

Supporting this, a study looked at speed dating, where men and women with no prior information about each other, conversed for 3 minutes, before choosing whether they desired further contact. The major factors determining positive choices were all physical, for both women and men. Women with lower BMI, younger age, and greater self-rated facial attractiveness generated more positive responses. Women’s height had no effect. For women age and positive partner responses had a highly statistically significant negative correlation.7 Similarly a study of online dating behavior showed that, controlling for social status, there was a strong revealed preference by men for younger women. In contrast women had a weak positive preference for older men.8 Consistent with this, male ratings of the attractiveness of female faces declined more with age than did female ratings of the attractiveness of male faces.9

7 Kurzban, Robert, and Jason Weeden, HurryDate: Mate preferences in action, Evolution and Human Behavior 26(3), 2005: 227-244. https://doi.org/10.1016/j.evolhumbehav.2004.08.012

8 Bruch, Elizabeth E. and M. E. J. Newman. Aspirational pursuit of mates in online dating markets. Science Advances 4, 2018, eaap9815. DOI:10.1126/sciadv.aap9815

9 Ebner, N. C. Age of face matters: Age-group differences in ratings of young and old faces. Behavior Research Methods, 40(1), 2008, 130–136.

- In the Methodology, you use CAMSIS 1990 scores of social status for marriages 1940-2021. For marriage 1837-1939 you have constructed their own occupational status association index. While you have responded partly to reviewers query about this, we don't feel the submission fully explains why two different indexes were used and why the cut-off of 1939/1940 was chosen.

our reply

In the text we have added the clarification:

We employ a separate occupational status index for 1837-1939 because over the long interval 1837-2021 many new occupations emerged, and the social status of some occupations changed significantly. Thus the 1837-1939 index shows a much higher father-son correlation of 0.71 1837-1879, compared to only 0.58 for the CAMSIS 1990 index for that period. Figure 2, showing an absence of significant hypergamy 1837-1939, would be unchanged if we instead measured father status 1837-1939 using the CAMSIS 1990 index.

We used the earlier occupational status index for 1837-1939 because we anticipated being criticized for using an inappropriate status index for the earlier years, had we used CAMSIS 1990 throughout. We chose the cutoff of 1940 because the period 1940-2021 saw much faster rates of technological change and economic growth than 1837-1939. 

- Reviewer 1 commented on how fathers' social status may not be the same as their children's social status, and further exploration of parental status vs child status vs married status would be helpful, and while your response to this was considered, we feel this limitation should be at least explicitly stated.

- Additional consideration and discussion of the role of maternal status would be helpful

our reply

In the paper we measure status by indicators of family status, either fathers’ occupational status, or surname status. We do this because for most of the period covered, women do not have a status of their own, but only their family status – birth family or husband. This provides a concrete and consistent measure of hypergamy: what is the family status at birth of women relative to their husband’s family status at birth?

Son and daughter status will regress to the mean compared to fathers’ status. But if that regression to the mean is the same for sons as for daughters, then measuring status using father status will not bias our findings. The only way this indirect measure of child status would be misleading would be if daughter status systematically declined relative to father status. We see no reason to expect this.

But to try and address this referee concern in the paper we have added the following paragraph in the discussion section.

As noted above, we consistently measure hypergamy relative to birth-family status of husband and wife. If daughters’ attained status systematically declined to relative birth-family social status, in a way that sons’ status did not, then there would be a form of hypergamy. But we find no evidence that brides in England 1837-2021 were regarded as having degraded status relative to their birth families. 

- Finally, the structure of the submission also remains unclear, and while the authors have made some changes related to this, we kindly request a clearer adherence to the PLOS ONE manuscript guidelines (https://journals.plos.org/plosone/s/submission-guidelines#loc-manuscript-organization).

The manuscript has been re-arranged using recent PLOS published papers also as a guide.

our reply

We have strictly conformed to the suggested pattern of

Abstract

Introduction

Methods

Results

Discussion

Acknowledgements

References

---

## [Editor Report · Decision Letter 2]

16 Dec 2024

Hypergamy Reconsidered: Marriage in England, 1837-2021

PONE-D-24-22853R2

Dear Dr. Cummins,

We’re pleased to inform you that your manuscript has been judged scientifically suitable for publication and will be formally accepted for publication once it meets all outstanding technical requirements.

Kind regards,

Grażyna Liczbińska

Academic Editor

PLOS ONE
---

## [Editor Report · Acceptance letter]

2 Jan 2025

PONE-D-24-22853R2 

PLOS ONE

Dear Dr. Cummins, 

I'm pleased to inform you that your manuscript has been deemed suitable for publication in PLOS ONE. Congratulations! Your manuscript is now being handed over to our production team.

Kind regards, 

on behalf of

Dr. Grażyna Liczbińska 

Academic Editor

PLOS ONE